# Neutrophile-Lymphocyte Ratio and Outcome in Takotsubo Syndrome

**DOI:** 10.3390/biology11081154

**Published:** 2022-08-01

**Authors:** David Zweiker, Edita Pogran, Laura Gargiulo, Ahmed Abd El-Razek, Ivan Lechner, Ivan Vosko, Stefan Rechberger, Heiko Bugger, Günter Christ, Diana Bonderman, Evelyn Kunschitz, Clara Czedik-Eysenberg, Antonia Roithinger, Valerie Weihs, Christoph C. Kaufmann, Andreas Zirlik, Axel Bauer, Bernhard Metzler, Thomas Lambert, Clemens Steinwender, Kurt Huber

**Affiliations:** 13rd Medical Department, Cardiology and Intensive Care Medicine, Clinic Ottakring (Wilhelminenhospital), 1160 Vienna, Austria; edita.pogran@gmail.com (E.P.); valerie.weihs@chello.at (V.W.); christoph.c.kaufmann@gmail.com (C.C.K.); kurt.huber@meduniwien.ac.at (K.H.); 2Department of Cardiology, Medical University of Graz, 8036 Graz, Austria; ivan.vosko@medunigraz.at (I.V.); heiko.bugger@medunigraz.at (H.B.); andreas.zirlik@medunigraz.at (A.Z.); 3Faculty of Medicine, Sigmund Freud University, 1020 Vienna, Austria; laura.gargiulo@web.de (L.G.); ahmed.razek@greatward.com (A.A.E.-R.); antonia.roithinger@gmx.at (A.R.); 4Clinic of Internal Medicine III—Cardiology and Angiology, Medical University of Innsbruck, 6020 Innsbruck, Austria; ivan.lechner@tirol-kliniken.at (I.L.); axel.bauer@tirol-kliniken.at (A.B.); bernhard.metzler@tirol-kliniken.at (B.M.); 5Department of Cardiology and Intensive Medicine, Kepler University Clinic, 4020 Linz, Austria; stefan.rechberger@kepleruniklinikum.at (S.R.); thomas.lambert@kepleruniklinikum.at (T.L.); clemens.steinwender@kepleruniklinikum.at (C.S.); 65th Medical Department for Cardiology, Clinic Favoriten, 1100 Vienna, Austria; guenter.christ@gesundheitsverbund.at (G.C.); diana.bonderman@gesundheitsverbund.at (D.B.); 72nd Medical Department, Hanusch Hospital, 1140 Vienna, Austria; evelyn.kunschitz@oegk.at; 8Core Facility, Medical University of Vienna, 1090 Vienna, Austria; clara.czedik-eysenberg@gmx.net

**Keywords:** neutrophile-lymphocyte ratio, takotsubo syndrome, predictors, registry, LVEF, outcome

## Abstract

**Simple Summary:**

Takotsubo syndrome, also known as “stress-induced cardiomyopathy” is a disease of the heart that occurs after physical or emotional stress and can mimic a heart attack. Patients with this disease may become critically ill with the need for treatment at the intensive care unit or may even die. For medical professionals it is very important to identify patients that worsen. Therefore, we performed an analysis of a large registry of patients with Takotsubo syndrome in Austria that includes 338 patients. As already known, patients with reduced cardiac function (measured by “left ventricular ejection fraction”) have worse outcomes. Additionally, we present the “neutrophile lymphocyte ratio” (NLR), which is a new parameter deriving from standard blood count parameters. NLR identifies patients at high risk of complications, early mortality and death at long-term follow up.

**Abstract:**

Background: Takotsubo syndrome (TTS) is an important type of acute heart failure with significant risk of acute complications and death. In this analysis we sought to identify predictors for in-hospital clinical outcome in TTS patients and present long-term outcomes. Methods: In this analysis from the Austrian national TTS registry, univariable and multivariable analyses were performed to identify significant predictors for severe in-hospital complications requiring immediate invasive treatment or leading to irreversible damage, such as cardiogenic shock, intubation, stroke, arrhythmias and death. Furthermore, the influence of independent predictors on long-term survival was evaluated. Results: A total of 338 patients (median age 72 years, 86.9% female) from six centers were included. Severe in-hospital complications occurred in 14.5% of patients. In multivariable analysis, high neutrophile-lymphocyte-ratio (NLR; OR 1.04 [95% CI 1.02–1.07], *p* = 0.009) and low LVEF (OR 0.92 [0.90–0.95] per %, *p* < 0.001) were significant predictors of severe in-hospital complications. Both the highest NLR tercile and the lowest LVEF tercile were significantly associated with reduced 5-year survival. Conclusions: Low LVEF and high NLR at admission were independently associated with increased in-hospital complications and reduced long-term survival in TTS patients. NLR is a new easy-to-measure tool to predict worse short- and long-term outcome after TTS.

## 1. Introduction

Takotsubo syndrome (TTS) is an acute reversible heart failure syndrome [1] that leads to symptoms similar to those of acute coronary syndrome (ACS). TTS is characterized by transient systolic and diastolic left ventricular dysfunction with various wall motion abnormalities with the exclusion of significant coronary artery disease [1,2]. While there are several theories of the underlying mechanisms of TTS, such as increased sympathetic activity, catecholamine surge and microvascular dysfunction, the exact pathophysiology of the condition is still unclear [3].

TTS was often thought to be associated with favorable outcomes compared to ACS [4]. However, large-scale analyses revealed that up to 50% of patients suffer from acute complications and acute mortality was similar to ACS (1.7–4.1%, [5,6,7]).

The identification of patients that may develop relevant complications during hospitalization is therefore of major clinical interest. Several factors have been identified as predictors of complications, such as male sex, high age and reduced left-ventricular function [8,9,10,11,12]. Recently, hematological indices, such as neutrophile to lymphocyte ratio (NLR), have been identified as possible predictors of in-hospital complications [13]. The goal of this study was the identification of solid predictors for short- and long-term outcomes in hospitalized patients with TTS from a multi-centric prospective registry.

## 2. Materials and Methods

This study is a retrospective analysis of the multicentric Austrian Takotsubo Syndrome Registry [14], which prospectively collects baseline characteristics and clinical outcome of patients with TTS at several centers in Austria. The registry is approved by the institutional review boards of the city of Vienna (application no. EK-21-071-VK), the Medical University of Graz (no. 33-084), the Medical University of Innsbruck (no. 1403/2020) and the Medical Faculty of the Johannes Kepler University Linz (no. 1276/2020). The investigation conforms with the principles outlined in the Declaration of Helsinki [15].

### 2.1. Study Population

Six Austrian hospitals participated in this study (3rd Medical Department for Cardiology and Intensive Care Medicine; Clinic Ottakring, Vienna, Austria; 5th Medical Department of Cardiology, Clinic Favoriten, Vienna, Austria; 2nd Medical Department, Hanusch Hospital, Vienna, Austria; University Clinic of Internal Medicine III, Cardiology and Angiology, Medical University of Innsbruck, Austria; Department for Cardiology, Medical University of Graz, Graz, Austria; Department of Cardiology and Intensive Medicine, Kepler University Clinic, Linz, Austria).

Main inclusion criteria were the European Heart Failure Association criteria for the diagnosis of TTS [1]: (1) Transient left ventricular dysfunction with wall motion abnormality, which extends beyond a single epicardial vascular distribution; (2) The absence of culprit atherosclerotic coronary artery disease including acute plaque rupture, thrombus formation, and coronary dissection; (3) New and reversible electrocardiography (ECG) abnormalities during the acute phase; (4) Significant elevation of serum natriuretic peptide (BNP or NT-proBNP) and relatively small elevation of cardiac troponin measured; and (5) Recovery of ventricular systolic function on cardiac imaging at follow-up (3–6 months), respectively.

Subjects with TTS-like presentation such as myocarditis, subarachnoid hemorrhage, pheochromocytoma, or hypertrophic cardiomyopathy were excluded from the database.

### 2.2. Data Collection

The database was constructed and completed according to the declaration of Helsinki [16]. Trained medical personnel at each center gathered the subject’s medical history, cardiovascular risk factors, laboratory values, echocardiography findings and in-hospital outcomes by reviewing the patient’s data records. The data were sent to the responsible study coordinator at Clinic Ottakring [17] for further processing and analysis.

Laboratory parameters at admission were defined as the values measured within the first 24 h after admission to the hospital. If there were several blood tests performed in the first 24 h, the highest value was included into the statistical analysis.

Complete long-term survival was verified and facilitated by data from the Austrian government’s population registry [18].

### 2.3. Endpoints

Severe in-hospital complications were all complications requiring immediate invasive therapy or leading to irreversible disability and included cardiogenic shock requiring catecholamines; respiratory insufficiency requiring intubation; ischemic stroke; ventricular fibrillation; persistent ventricular tachycardia; symptomatic high-grade atrioventricular block; and intrahospital death, respectively. Other complications were considered non-severe. If more than one complication occurred in a patient, only complications considered most severe were documented.

### 2.4. Statistics

Statistics was performed with R version 4.1.2 (The R Project for Statistical Computing, Vienna, Austria) and RStudio version 2021.09.1 + 372 (RStudio Inc, Boston, MA, USA). Data are presented in mean ± standard deviation, median (interquartile range) or proportion (absolute number), where appropriate. The primary endpoint was the occurrence of any severe in-hospital complication or death. All parameters were available in >90% of patients, unless declared otherwise in Table 1. Univariable analysis was performed using Fisher exact test, Student’s *t* test or Wilcoxon rank-sum test, where appropriate. Various Troponin test kits were used in different centers; accordingly, elevated cardiac troponin (defined as above the individual threshold, normally the 99th percentile) was included in the univariable analysis. The GEIST score was calculated according to Santoro et al. [10]. To compensate for missing data in multivariable analysis, multiple imputation with five iterations using the package *mice* was performed. Logistic regression was performed with generalized linear modelling using the *glm* function, including all significant predictors (*p* < 0.05) from univariable analysis and age, excluding right-ventricular function and the GEIST score due to missing data in >50% of patients. Independent predictors for short-term complications were also analyzed regarding their association with long-term survival, with stratification into terciles, if applicable. Furthermore, a subgroup analysis with stratification by age, sex and left-ventricular function was performed.

## 3. Results

During the observation period, a total of 338 patients from six centers were included. Median age was 72 years and 86.9% were female. The most common co-morbidities were arterial hypertension (61.7%), hyperlipidemia (34.1%) and COPD (22.0%, Table 1). An underlying psychiatric disorder was present in 21.4% of TTS patients. Physical triggers were present in 25.4%, emotional triggers alone in 21.4%. The remaining patients had either or both physical and emotional triggers (4.3%), or no triggers at all (49.1%).

**Table 1 biology-11-01154-t001:** Baseline characteristics of the total study population and stratified by subgroups based on occurrence of severe in-hospital complications. Severe in-hospital complications were all complications requiring immediate invasive therapy or leading to irreversible disability.

Parameter	Total	No Complications (*n* = 288)	Complications (*n* = 49)	*p* Value
**demographics**				
female sex	86.9% (*n* = 293)	87.5% (*n* = 253)	83.3% (*n* = 40)	0.486
age	72 (62–79)	72 (62–79)	73 (65–81)	0.437
**comorbidities**				
arterial hypertension	61.7% (*n* = 208)	61.5% (*n* = 177)	63.3% (*n* = 31)	0.875
hyperlipidemia	34.1% (*n* = 115)	34.7% (*n* = 100)	30.6% (*n* = 15)	0.628
history of coronary artery disease	11.0% (*n* = 37)	10.8% (*n* = 31)	12.2% (*n* = 6)	0.805
atrial fibrillation	12.8% (*n* = 43)	12.5% (*n* = 36)	14.3% (*n* = 7)	0.816
other supraventricular arrhythmia	1.2% (*n* = 4)	1.4% (*n* = 4)	0% (*n* = 0)	1.000
chronic kidney disease	13.6% (*n* = 46)	10.4% (*n* = 30)	32.7% (*n* = 16)	<0.001 *
COPD	22% (*n* = 74)	21.9% (*n* = 63)	22.4% (*n* = 11)	1.000
diabetes mellitus type 2	15.7% (*n* = 53)	14.6% (*n* = 42)	22.4% (*n* = 11)	0.201
psychiatric disease	21.4% (*n* = 72)	22.9% (*n* = 66)	12.2% (*n* = 6)	0.130
history of previous TTS	1.2% (*n* = 4)	1.4% (*n* = 4)	0% (*n* = 0)	1.000
current smoker	23.5% (*n* = 74)	22.2% (*n* = 59)	30.6% (*n* = 15)	0.204
previous smoker	14.3% (*n* = 45)	16.5% (*n* = 44)	2% (*n* = 1)	0.006 *
**triggers for TTS**				
emotional trigger	21.9% (*n* = 74)	24.2% (*n* = 70)	8.2% (*n* = 4)	0.014 *
physical trigger	25.4% (*n* = 86)	24.2% (*n* = 70)	32.7% (*n* = 16)	0.217
both emotional and physical trigger	4.3% (*n* = 12)	4.3% (*n* = 10)	4.9% (*n* = 2)	0.695
unclear trigger	49.1% (*n* = 166)	48.1% (*n* = 139)	55.1% (*n* = 27)	0.440
**laboratory tests**				
hs-troponin T (ng/L, *n* = 121 [35.8%])	192 (69–416)	194 (75–417.75)	138 (53–328)	0.361
hs-troponin I (pg/mL, *n* = 7 [2.1%])	3165 ± 3726	3165 ± 3726	N/A	N/A
troponin I (ng/mL, *n* = 195 [57.7%])	1.82 (0.448–4.105)	1.59 (0.448–4.023)	2.215 (0.458–4.803)	0.603
elevated troponin	97.7% (*n* = 214)	97.3% (*n* = 177)	100% (*n* = 37)	0.592
leukocytes at admission (G/L)	9.88 (7.82–13.17)	9.90 (7.81–12.99)	9.61 (7.85–15.51)	0.622
CRP at admission (mg/L)	5.4 (2.8–14.3)	4.8 (2.8–12.6)	9.2 (4.1–60.4)	0.002 *
neutrophile granulocytes (G/L)	7.3 (5.0–10.9)	6.9 (5.0–10.4)	9.3 (6.3–15.1)	0.005 *
lymphocytes (G/L)	1.7 (1.11–2.32)	1.8 (1.2–2.4)	1.5 (0.9–2.1)	0.019 *
NLR at admission	4.33 (2.51–7.93)	4.05 (2.49–7.00)	6.12 (3.40–15.09)	0.002 *
nt-proBNP (pg/mL, *n* = 198 [58.6%])	2352 (742–6030)	2417 (738–5799)	1930 (784–7739)	0.828
**imaging**				
left ventricular ejection fraction (%, *n* = 285 [84.3%])	46.0 ± 13.7	48.0 ± 12.3	35.4 ± 15.4	<0.001 *
right ventricular involvement (*n* = 167 [49.4%]	5.4% (*n* = 9)	5.0% (*n* = 7)	7.4% (*n* = 2)	0.639
apical ballooning	48.5% (*n* = 148)	51.7% (*n* = 134)	30.4% (*n* = 14)	0.010 *
midventricular ballooning	48.5% (*n* = 148)	45.2% (*n* = 117)	67.4% (*n* = 31)	0.006 *
basal ballooning	2.3% (*n* = 7)	2.3% (*n* = 6)	2.2% (*n* = 1)	1.000
**combined scores**				
GEIST score [10] (*n* = 161 [47.6%])	−0.4 (−0.5–19.5)	−0.4 (−0.5–19.5)	−0.2 (−0.4–19.6)	0.034 *
**hospital stay**				
hospital stay (days)	5 (3–10)	5 (3–8)	11 (6–25)	<0.001 *

CRP: C-reactive protein; COPD: chronic obstructive pulmonary disease; GEIST score: chronic kidney disease: eGFR < 60 mL/min at ≥2 measurements before the TTS event; NLR: neutrophile/lymphocyte ratio; TTS: Takotsubo syndrome. * *p* < 0.05.

Laboratory tests at admission showed elevated median troponin and Nt-proBNP levels (Table 1). The median NLR was 4.3. The median leukocyte count was 9.88 G/L. Mean LVEF was 46.5% with apical wall motion abnormalities being present in almost half of patients (48.5%). In-hospital complications occurred in 33.1% (Table 2), with a range from 0% to 47% in each center, whereas 14.5% were classified as severe (Figure 1). Most prevalent complications were cardiogenic shock (9.8%), respiratory failure without intubation (7.7%), acute kidney failure (5.3%) and bleeding (5.0%). In-hospital death occurred in 3.3%. Median (IQR) hospital stay was 5 (3–10) days.

In univariable analysis, chronic kidney disease and the absence of emotional events prior to the development of TTS were significant predictors for in-hospital complications (Table 1). Interestingly, previous smoking was associated with a lower complication rate compared to current smoking or no smoking. Patients with in-hospital complications had significantly higher C-reactive protein (CRP) concentrations and NLR values at admission, as well as lower LVEF (*p* < 0.05 for each, Table 1). High NLR in patients with complications was driven by both significantly higher neutrophile granulocyte count and lower lymphocyte count (Table 1). Apical ballooning was significantly less prevalent in patients with complications (30.4 vs. 51.7%, *p* = 0.010) while midventricular ballooning was more prevalent (67.4% vs. 45.2%, *p* = 0.006). As expectable, in-hospital complications were associated with a significantly longer hospital stay (median [IQR] 11 [6,7,8,9,10,11,12,13,14,15,16,17,18,19,20,21,22,23,24,25] vs. 5 [3,4,5,6,7,8] days, *p* < 0.001).

In multivariable regression analysis, high NLR at admission and a reduced LVEF remained significant predictors of in-hospital complications. When comparing upper and lower terciles, high NLR was associated with a significantly higher risk of total complications, severe complications and in-hospital death (*p* < 0.005 for all, Figure 2, Table 3). The predictive value of NLR was similar in subgroups stratified by age and sex, with significant interaction of left-ventricular function only (Table A1).

Mean long-term follow-up was 3.2 ± 3.5 years. Overall cumulative five-year survival was 70.2%. Five-year survival was significantly lower in the patient group with the highest NLR tercile (59.3% vs. 2nd tercile 75.0% vs. 1st tercile 76.9%, Log-rank *p* = 0.007, Figure 3) and lowest LVEF tercile (51.2% vs. 2nd tercile 77.3% vs. 3rd tercile 75.5%, *p* < 0.001, Figure A1).

## 4. Discussion

This multicentric study including 338 patients with TTS demonstrates that: (1) severe in-hospital complications occur in a considerable proportion of patients with TTS (14.5%); (2) additional to low LVEF, high NLR was a significant predictor of severe in-hospital complications; and (3) high NLR was also associated with reduced long-term survival.

Similar to previous studies, the majority of patients presenting with TTS were postmenopausal women with a prevalence of comorbidities similar to the overall population, such as arterial hypertension (61.7%) [19,20]. However, the rate of patients with psychiatric disease was high (21.4%), which is distinctive of a TTS population [21,22,23,24].

The high rate of in-hospital complications (33.1%) confirms that TTS is a serious disease. Roughly half of those complications (14.5% of the total population) were considered severe, i.e., required immediate treatment or led to irreversible disability. In-hospital mortality was 3.3%. These results are in line with previous recent literature that present a complication rate of 23–40% [10,13,25,26,27] and in-hospital mortality from 3% to 13% [8,9,10,11,12,13,25,26,27,28,29,30].

In univariable analysis, we found classical predictors for in-hospital complications, such as low LVEF [8,12,25,27], chronic kidney disease [27] and absence of emotional trigger events [25], similar to existing literature. The GEIST score [10], which was available for a fraction of included patients, was also predictive of in-hospital complications. In concordance with Uribarri et al. [31], the region of wall motion abnormalities in echocardiography or fluoroscopy was associated with acute outcome, favoring the apical type of TTS. Additionally, high NLR was correlated with poor outcome. One can only speculate why patients who stopped smoking earlier in their life had a better outcome than the remaining patients in the univariable analysis. If this effect persists in other studies, further research is needed to elucidate the pathophysiological association between smoking and short-term outcome after TTS.

Low LVEF and high NLR were the only independent predictors of severe in-hospital complications in multivariable analysis. Furthermore, long-term survival was reduced in patients with high NLR and low LVEF. While low LVEF was a consistent independent predictor in various studies [8,10,12,25,27], the predictive value of NLR has not been identified until recently in TTS [13].

NLR as an inflammatory marker has been identified as an outcome predictor for several cardiovascular diseases [32]. The interplay between inflammation and outcome is already well established in coronary artery disease [33]. In ACS, high NLR has been associated with poor outcome [34,35]. Furthermore, NLR is associated with the risk of AF development after coronary arterial bypass surgery and radiofrequency ablation [36,37,38]. High NLR also seems to be predictive of worse outcome in patients with heart failure [39] and after valvular interventions [40].

The exact pathophysiological association between NLR and acute outcome is still not clear in TTS patients. As neutrophils are the first cells found in necrotic myocardial tissue [34], they may be a direct sign of the amount of damaged myocardium. This effect is underlined by the association of high absolute white blood count with worse long-term outcomes [41]. In TTS, which is mainly induced by sympathetic activation [1,42], effects of catecholamines on white cell counts, promoting neutrophilia and homing of lymphocytes [43] may play an even more important role. Catecholamines might potentiate systemic inflammatory response particularly in the setting of infections and tissue injury [43,44,45]. Since infections and injuries belong to physical triggers of TTS, this may be an explanation for the absence of emotional triggers to be an independent predictor of in-hospital complications in univariable analysis.

Furthermore, the study by Scally et al. demonstrated that TTS is characterized by a myocardial macrophage inflammatory infiltrate and an increase in systemic inflammatory cytokines. Pro-inflammatory cytokines persisted for at least 5 months, suggesting a low chronic inflammatory state [46]. Additionally, a constant myocardial oedema with adjuvant reduction in myocardial phosphorus-creatinine ratio as an indicator for energetic impairment was shown by using magnetic resonance imaging [46].

Our results demonstrated that high CRP as an acute phase protein was associated with a worse outcome in univariable analysis. This finding agrees with findings from Lachmet-Thebaud et al., who showed that a residual high-inflammatory response, defined as CRP > 19 mg/L, was an independent predictor of cardiovascular death and heart failure in TTS [47]. However, NLR outperformed CRP as a predictive marker in this analysis. To the authors’ knowledge, no other study has yet compared the predictive value of NLR vs. CRP in TTS patients. Nevertheless, these findings underline the importance of inflammation in TTS. Hence, further clinical research is required to consider whether the inflammation might be used as a therapeutic target.

Moreover, NLR can be calculated easily from a regular hemogram. Therefore, it may be used to identify patients already at risk early during hospital admission, in addition to existing clinical parameters. Besides single parameters (such as LVEF), Santoro et al. suggested the GEIST prognostic score (including LVEF, right-ventricular involvement, sex and neurologic disorders) to find patients with an increased risk for in-hospital complications [10]. This score was predictive of in-hospital complications in a fraction of our population. Unfortunately, we were not able to validate this risk score in the complete population due to missing data. It is currently unclear whether the NLR has predictive value on top of the GEIST score.

For the use of NLR in daily clinical practice, a specific cut-off value identifying patients with increased risk is necessary. Currently, no such threshold is available, and previous literature used different cut-offs: Santoro et al. found increased complications in patients with an NLR > 5 [13], while Dominguez et al. used NLR > 4 in their transcatheter aortic valve implantation cohort [40]. For risk stratification, we stratified patients in terciles, similarly to Benites-Zapata et al. in their heart failure cohort [39], and came to a similar threshold (5.92 vs. Benites-Zapata et al., 5.4).

### Limitations and Strengths

Despite the large sample size of consecutive patients and the multicentric design with a high data quality, the results have to be taken with care as the risk of several biases exists in retrospective analyses. Furthermore, the data quality was impaired in a proportion of patients due to missing values, especially concerning right-ventricular involvement.

## 5. Conclusions

This analysis shows that patients with TTS have a high rate of in-hospital complications and patients at risk may be identified by high NLR at admission. NLR may have a predictive value on top of existing risk scores for both short and long-term outcomes.

## Figures and Tables

**Figure 1 biology-11-01154-f001:**
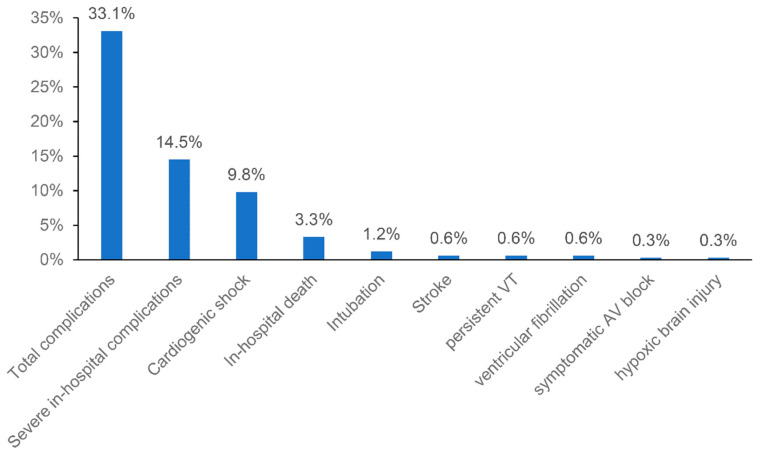
Total and severe in-hospital complications in the whole study population.

**Figure 2 biology-11-01154-f002:**
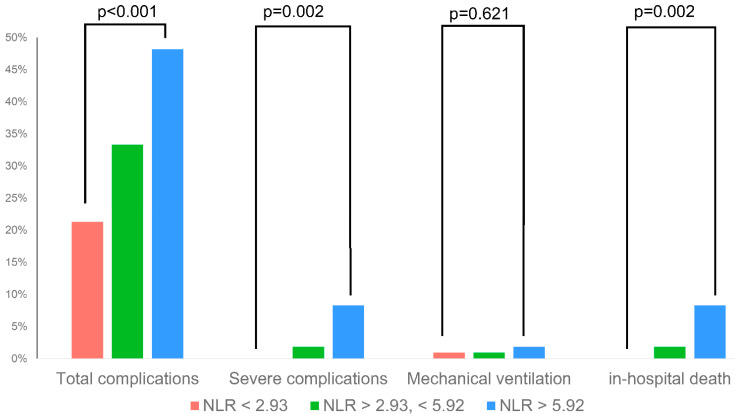
Rate of selected in-hospital complications across different NLR terciles.

**Figure 3 biology-11-01154-f003:**
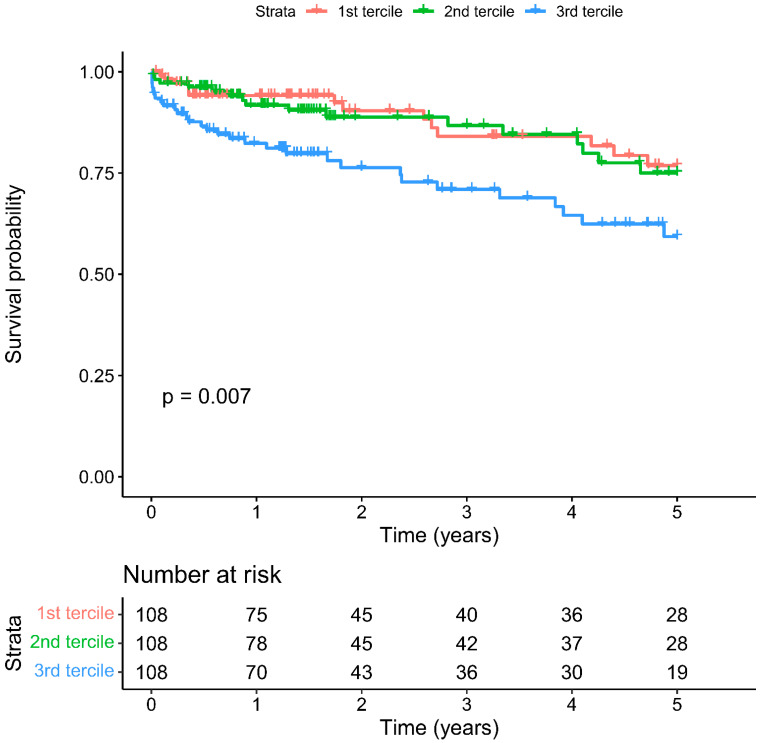
Cumulative five-year mortality in NLR terciles.

**Table 2 biology-11-01154-t002:** Total, severe and other in-hospital complications in the whole study population.

Complication	Proportion (Absolute Count)
**total in-hospital complications**	33.1% (*n* = 112)
**severe in-hospital complications**	14.5% (*n* = 49)
cardiogenic shock	9.8% (*n* = 33)
in-hospital death	3.3% (*n* = 11)
respiratory failure requiring intubation	1.2% (*n* = 4)
stroke	0.6% (*n* = 2)
persistent ventricular tachycardia	0.6% (*n* = 2)
ventricular fibrillation	0.6% (*n* = 2)
symptomatic AV block	0.3% (*n* = 1)
hypoxic brain injury	0.3% (*n* = 1)
**other in-hospital complications**	
respiratory failure without intubation	7.7% (*n* = 26)
new-onset atrial fibrillation	3.6% (*n* = 12)
acute kidney failure	5.3% (*n* = 18)
Bleeding	5.0% (*n* = 17)
atrial fibrillation	3.6% (*n* = 12)
urinary tract infection	3.6% (*n* = 12)
pneumonia	2.7% (*n* = 9)
sepsis (without shock)	1.5% (*n* = 5)
left-ventricular thrombus	0.9% (*n* = 3)
mitral regurgitation (conservative treatment)	1.2% (*n* = 4)
infection of unknown origin	0.9% (*n* = 3)
pericardial effusion	0.3% (*n* = 1)
enteritis	0.3% (*n* = 1)
lower limb ischemia	0.3% (*n* = 1)
aneurysma spurium	0.3% (*n* = 1)

AV: atrioventricular.

**Table 3 biology-11-01154-t003:** Univariable and multivariable predictors of in-hospital complications.

	Univariable Analysis		Multivariable Analysis	
Parameter	OR (95% CI)	*p* Value	OR (95% CI)	*p* Value
age (per year)	1.01 (0.98–1.04)	0.437	1.00 (0.97–1.04)	0.822
chronic kidney disease	4.18 (2.03–8.41)	<0.001	1.84 (0.78–4.37)	0.164
previous smoker	0.11 (0.01–0.50)	0.006	0.13 (0.02–1.06)	0.056
emotional trigger	0.28 (0.08–0.72)	0.014	0.55 (0.17–1.76)	0.312
CRP at admission	1.01 (1.00–1.02)	0.002	1.01 (1.00–1.02)	0.076
NLR at admission	1.04 (1.02–1.07)	0.002	1.04 (1.01–1.08)	0.009 *
LVEF (per %)	0.92 (0.90–0.95)	<0.001	0.93 (0.90–0.96)	<0.001 *
apical ballooning	0.41 (0.20–0.79)	0.010	0.78 (0.08–7.50)	0.826
midventricular ballooning	2.51 (1.31–4.99)	0.006	0.98 (0.10–9.84)	0.986

* *p* < 0.05 in multivariable analysis.

## Data Availability

Original data supporting reported results are available on request from the corresponding author.

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
