# Peer review of "Neutrophile-Lymphocyte Ratio and Outcome in Takotsubo Syndrome"

_biology, 2022, doi:10.3390/biology11081154_

Round 1

Reviewer 1 Report

Dear Authors,

Overall, well-documented paper on a challenging issue, with expected clinical impact.

Major Comments:

According to Afari et al (ref. 32): "Also, one of the biggest challenges will be to determine a clinically relevant cut-off NLR value.".

In this regard, it would be interesting to include a comparison of the analysed Takotsubo NLR values with relevant literature reports.

Minor comments:

Line 45: "...predictors with long-term..."                                                           

Suggested amendment: "...predictors on long-term..."

Line 252: "...phosphor-creatinine ratio as an indicator for energetic impairment was shown by using a magnetic resonance imaging [46]."

Suggested amendment: "...phosphorus-creatinine ratio as an indicator for energetic impairment was shown by using magnetic resonance imaging [46]."

Author Response

Major Comments:
According to Afari et al (ref. 32): "Also, one of the biggest challenges will be to determine a clinically relevant cut-off NLR value.".
In this regard, it would be interesting to include a comparison of the analysed Takotsubo NLR values with relevant literature reports.

  • The absence of a useful cut-off for NLR is a major drawback of this promising parameter. We added a paragraph about this topic in the Discussion at lines 277-283.

Minor comments:
Line 45: "...predictors with long-term
Suggested amendment: "...predictors on long-term..."
Line 252: "...phosphor-creatinine ratio as an indicator for energetic impairment was shown by using a magnetic resonance imaging [46]."
Suggested amendment: "...phosphorus-creatinine ratio as an indicator for energetic impairment was shown by using magnetic resonance imaging [46]."

  • We thank the reviewer for the minor comments and adjusted the manuscript accordingly.

Reviewer 2 Report

The aim of this manuscript is to study the relationship between neutrophile-lymphocyte ratio (NLR) and clinic outcome in patients with Takotsubo syndrome by a multi-center retrospective analysis. They found that elevated NLR could become a new predictor for the outcome in hospitalized patients with Takotsubo syndrome. This is overall very interesting retrospective study with statistic significant findings.

1.   1.      Figure legend and description of the Table should be included with more details.

2.   2.   In Table 1, please define the complications VS No complications.

3.   3.   Line 156, what do you mean by elevated Troponin,

4.   4.  Line 170-172 (need reference Table 3)

5.   5.   Line 170, “absence of emotional trigger” ?

6.   6.   Any explanation of why in Table 3, LVEF showed negative correlation with hospital complication?

7.   7.   In Table 3, NLR at admission, although showed statistic significant in terms of OR, but OR of 1.04 might have minimal clinical significance. Please explain. 

Author Response

The aim of this manuscript is to study the relationship between neutrophile-lymphocyte ratio (NLR) and clinic outcome in patients with Takotsubo syndrome by a multi-center retrospective analysis. They found that elevated NLR could become a new predictor for the outcome in hospitalized patients with Takotsubo syndrome. This is overall very interesting retrospective study with statistic significant findings.

  • We thank the reviewer for this positive comment.
  1.  Figure legend and description of the Table should be included with more details.
  • We adjusted all table legends and the legend of Figure 1 accordingly, including all abbreviations and a detailed description of the study population.
  1.  In Table 1, please define the complications VS No complications.
  • We defined severe in-hospital complications in the title of Table 1 accordingly.
  1.  Line 156, what do you mean by elevated Troponin,
  • As various different troponin kits were used in different centres (high-sensitive troponin T, high-sensitive troponin I, sensitive troponin I), the individual values were not comparable between centres. Therefore, we chose to include “elevated troponin”, defined as elevated value above the individual threshold of the centre, as parameter. Normally, elevated troponin values are defined above the 99th We adapted the sentence explaining this limitation in the Statistics section of the Methods (lines 132-134): “Various Troponin test kits were used in different centres; accordingly, elevated cardiac troponin (defined as above the individual threshold, normally the 99th percentile) was included into the univariable analysis.”
  1.  Line 170-172 (need reference Table 3)
  • We thank the author for this comment. However, as this text refers to univariable analysis, which is presented in Table 1, we added “(Table 1).” to the sentence.
  1.  Line 170, “absence of emotional trigger” ?
  • In the majority of patients with TTS, the disease develops after an event (“trigger”) of physical or emotional stress. In our study, patients without an emotional event had a higher risk of major in-hospital complications. For clarification, we changed the description to “emotional events prior to the development of TTS”.
  1.  Any explanation of why in Table 3, LVEF showed negative correlation with hospital complication?
  • Similar to other studies, such as Del Buono, M. G., et al. (2021), Jesel, L., et al. (2018) and Bento, D., et al. (2019), we found that reduced LVEF was associated with worse prognosis in TTS. As patients with higher LVEF values have less complications and those with lower LVEF have a higher risk of complications, LVEF was negatively correlated with hospital complications.
  1.  In Table 3, NLR at admission, although showed statistic significant in terms of OR, but OR of 1.04 might have minimal clinical significance. Please explain. 
  • As correctly pointed out by the reviewer, the OR for in-hospital complications per each increase of NLR is quite low, comparable to LVEF. Therefore, this parameter may not be suitable to identify patients at risk alone. However, as NLR is an independent predictor of worse outcomes, it may help increase the sensitivity and specificity of existing scores or already available clinical parameters. It would be very interesting examine the predictive value of the GEIST score, but due to missing data we were unable to do so in our complete population. To emphasize NLR especially to be used on top of existing parameter, we changed the Discussion as follows (Line 267-269): “Therefore, it may be used to identify patients at risk already early during hospital admission, in addition to existing clinical parameters.”